# Evidence for the Fulde–Ferrell–Larkin–Ovchinnikov state in bulk NbS$_2$

Chang-woo Cho [1,10], Jian Lyu [1,2,10], Cheuk Yin Ng[1,10], James Jun He [3,10], Kwan To Lo [1], Dmitriy Chareev[4,5,6], Tarob A. Abdel-Baset[7,8], Mahmoud Abdel-Hafiez[8,9] & Rolf Lortz [1✉]

We present measurements of the magnetic torque, specific heat and thermal expansion of the bulk transition metal dichalcogenide (TMD) superconductor NbS$_2$ in high magnetic fields, with its layer structure aligned strictly parallel to the field using a piezo rotary positioner. The upper critical field of superconducting TMDs in the 2D form is known to be dramatically enhanced by a special form of Ising spin orbit coupling. This Ising superconductivity is very robust to the Pauli paramagnetic effect and can therefore exist beyond the Pauli limit for superconductivity. We find that superconductivity beyond the Pauli limit still exists in bulk single crystals of NbS$_2$ for a precisely parallel field alignment. However, the comparison of our upper critical field transition line with numerical simulations rather points to the development of a Fulde-Ferrell-Larkin-Ovchinnikov state above the Pauli limit as a cause. This is also consistent with the observation of a magnetic field driven phase transition in the thermo-dynamic quantities within the superconducting state near the Pauli limit.

[1] Department of Physics, The Hong Kong University of Science and Technology, Clear Water Bay, Kowloon, Hong Kong. [2] Department of Physics, Southern University of Science and Technology, Xueyuan Road, Nanshan District, Shenzhen, Guangdong Province, China. [3] RIKEN Center for Emergent Matter Science (CEMS), Saitama, Wako, Japan. [4] Institute of Experimental Mineralogy of Russian Academy of Sciences, Chernogolovka, Moscow region, Russia. [5] National University of Science and Technology "MISiS", Moscow 119049, Russia. [6] Ural Federal University, Ekaterinburg 620002, Russia. [7] Department of Physics, Faculty of Science, Taibah University, Yanbu, Saudi Arabia. [8] Department of Physics, Faculty of Science, Fayoum University, Fayoum, Egypt. [9] Department of Physics and Astronomy, Uppsala University, Uppsala, Sweden. [10] These authors contributed equally: Chang-woo Cho, Jian Lyu, Cheuk Yin Ng, James Jun He. ✉email: lortz@ust.hk

Transition metal dichalcogenides have been the focus of recent research due to their wide range of unique electronic properties in their 2D form with high potential for technological applications[1–8]. Among them are intrinsic superconductors, such as 2H-NbSe$_2$ and 2H-NbS$_2$. In their 2D form, they have aroused great interest due to the discovery of Ising superconductivity, which allows them to exceed the Pauli limit of superconductivity[4,5]. While the upper critical field ($H_{c2}$) of a spin-singlet type-II superconductor is normally determined by the orbital limit for superconductivity[9], there are rare cases where the orbital limit is particularly high. Possible reasons are heavy effective masses of the quasiparticles[10–12], or a highly anisotropic structure that suppresses the orbital limit due to the open nature of the Fermi surface[13–24]. In this case, superconductivity is in principle abruptly destroyed at the Pauli limit[25,26] in the form of a first-order phase transition. There are two ways in which a superconductor can maintain its superconducting (SC) state above the Pauli limit. The first possibility is the formation of a Fulde–Ferrell–Larkin–Ovchinnikov (FFLO) state[27,28], in which the Cooper pairs obtain a finite center-of-mass momentum, resulting in a spatially modulated order parameter in a wide range of their magnetic field vs. temperature phase diagram, which can extend far beyond the Pauli limit. Such spatial modulation of superconductivity has been described as a pair density wave state and is also used to explain the pseudogap phase in cuprates[29]. Prominent examples of the FFLO state have been found in some layered organic superconductors[13–23], but also in form of the prominent Q-phase in the heavy-fermion compound CeCoIn$_5$, where a spatially modulated order parameter coexists with a magnetic spin density wave order[10–12], and more recently in the iron-based superconductor KFe$_2$As$_2$[24], and FeSe[30]. It is generally assumed that the superconductor must be in the clean limit for the FFLO state to be realized, with a mean-free path $\ell$ longer than the coherence length $\xi$[31]. Given the high purity of contemporary dichalcogenide samples and very short $\xi$ values, this criterion is certainly achieved here.

For 2D dichalcogenides, Ising superconductivity is another possibility[4,5]. Here, the breaking of the mirror symmetry in the plane leads to a very strong pinning of the electron spins out of the plane due to the Ising-Spin-Orbit interaction (ISOI), resulting in opposite spin directions in the adjacent K and K' electron pockets. This ISOI effectively protects the Cooper pairs when an in-plane magnetic field is applied to the 2D layer, resulting in enormous enhancements of the critical field. This phase is of particular interest due to the recent theoretical prediction of a topological SC state with Majorana zero modes in high parallel applied fields in monolayer NbSe$_2$[32,33].

While reports on Ising superconductivity usually focus on monolayers, it is also clear that $H_{c2}$ in 2D materials with multiple layers still exceeds the Pauli limit[5]. It is not clear whether ISOI could still have an influence in the bulk form of these layered materials with very weak interlayer coupling. In this letter, we present magnetic torque, specific heat, and thermal expansion measurements on a bulk single crystal of NbS$_2$, where we align the layered structure parallel to the applied magnetic field using a piezo rotary stage with millidegree accuracy. We find that $H_{c2}$ significantly exceeds the Pauli limit of 10 T at low temperatures and shows a pronounced characteristic upswing towards low temperatures. This is a clear indication that an unusual SC state is formed[14,15,24,34]. Thermal expansion measurements as a bulk thermodynamic method, which are closely related to the specific heat, indicate a small transition anomaly near the Pauli limit indicating a phase transition within the SC state. Using theoretical simulations[35], we show that Ising superconductivity is not accountable for such a strong $H_{c2}$ upturn in bulk NbS$_2$. The

existence of the additional phase transition within the SC state points to a formation of the FFLO state.

## Results

**Magnetic torque experiments.** Figure 1a shows magnetic torque magnitude ($\tau$) measured at various fixed temperatures with a small 1° misalignment of the field with respect to the basal plane of NbS$_2$. At lower temperatures, when $H_{c2}$ reaches higher fields, it is obvious that the transition becomes sharper and more step-like, which may be an indication for Pauli-limited first-order behavior[15,24]. Only a small tail persists above the step-shaped transition, which indicates a certain persistence of superconductivity. $H_{c2}$ reaches a maximum at 1.15 K and then decreases again slightly towards 0.35 K. Here we use the fields where the steepest slope occurs to mark the transition fields that should best represent the abrupt first-order-like transition. Note that other criteria (e.g., the transition onset) only lead to an almost parallel shift of the $H_{c2}$ line. To achieve a strictly parallel field orientation, we aligned the layered structure of the NbS$_2$ single crystal by first minimizing $\tau$ at a fixed field and temperature by gradually rotating the sample through the parallel orientation[24]. This allowed us to determine the approximate parallel orientation. Subsequently, we repeated field scans at tiny variations of orientation of 0.1° or less until we found torque data with minimum amplitude and minimal opening of the hysteresis loop. This corresponded to the parallel orientation. A sequence of measurements at small angular variation is shown in Fig. 1b. As the field decreases, the torque builds up much more continuously below $H_{c2}$, suggesting that the sharp jumps in Fig. 1a are due to screening currents that build up continuously during the field sweep in the SC state and decay abruptly when approaching the Pauli limit in slightly tilted fields.

In Fig. 2a, we show $\tau$ data measured at different fixed temperatures during field sweeps for the field applied parallel to the NbS$_2$ basal plane. Here we identify the $H_{c2}$ transition from the onset point where the data starts to deviate from zero, as marked

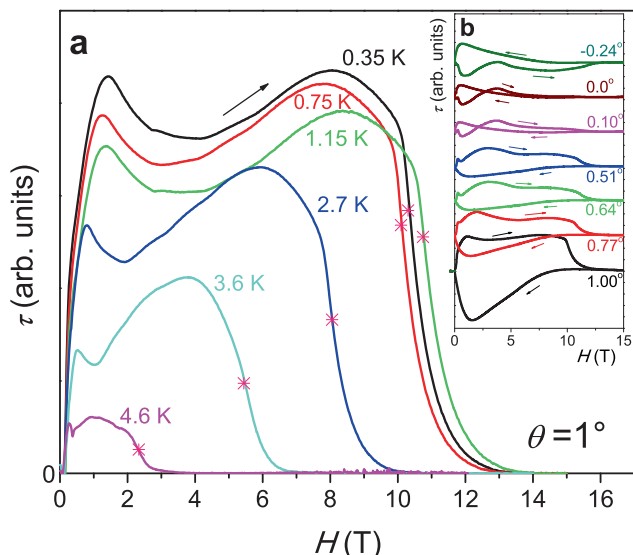

**Fig. 1 Magnetic torque data in a slightly tilted field of NbS$_2$. a** Magnetic torque magnitude $\tau$ ($H$) of NbS$_2$, measured at fixed temperatures as a function of the applied magnetic field $H$, oriented with a small angle $\theta = 1°$ to the basal plane. The data were measured at increasing field. Crosses mark the points where the steepest slope occurs near $H_{c2}$. **b** Magnetic torque measured at $T = 0.35$ K at various angles near $\theta = 0°$ (parallel field). A weak linear normal state contribution was removed from all data and offsets were added for better clarity.

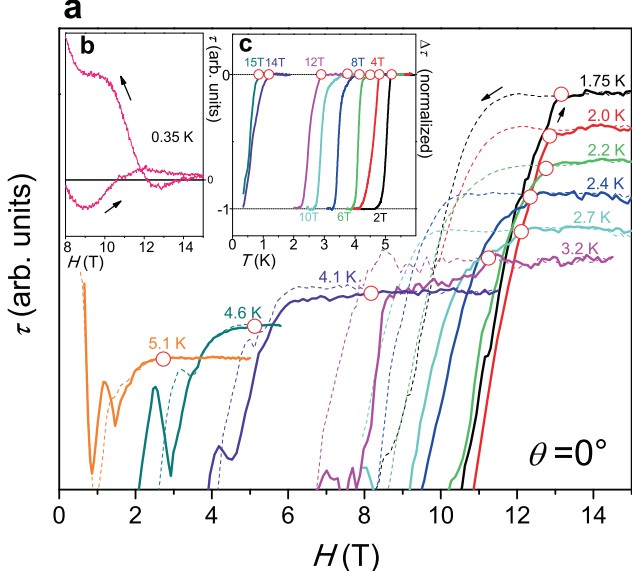

**Fig. 2 Magnetic torque data in parallel fields of NbS$_2$. a** Magnetic torque magnitude $\tau$ ($H$) measured at fixed temperatures as a function of the magnetic field applied strictly parallel to the basal plane ($\theta = 0°$) near the onset upper critical field $H_{c2}$ (open circles). The solid lines represent data measured at increasing field, while the dotted lines are data measured at decreasing field (overlapping parts of the data for low fields have been omitted and offsets have been added for reasons of clarity). **b** Magnetic torque data at 0.35 K where $H_{c2}$ exceeded the highest field in our magnet cryostat. **c** Magnetic torque measured in fixed parallel fields as a function of temperature. The data were normalized and a weak linear normal state background was removed for clarity. Open circles mark the onset critical temperatures.

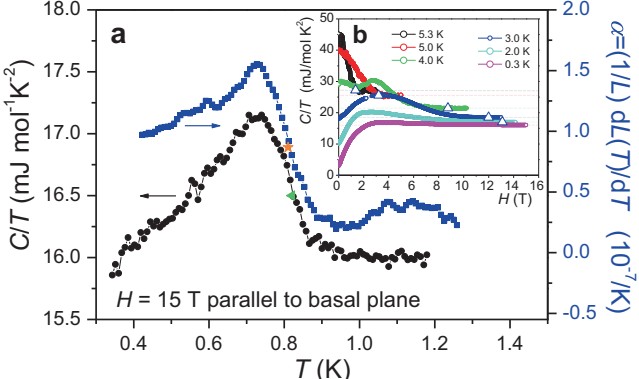

**Fig. 3 Thermodynamic bulk probes displaying the upper critical field $H_{c2}$ transition of NbS$_2$ in high parallel magnetic fields. a** Specific heat $C/T$ (circles) and linear thermal expansion coefficient $\alpha = (1/L)dL(T)/dT$ (squares) of NbS$_2$ showing the superconducting transition in a magnetic field of 15 T applied strictly parallel to the layer structure. The additional orange star and tilted green triangle mark the transition midpoints to be included in the phase diagram in Fig. 5. **b** Specific heat of NbS$_2$ measured with an *ac* technique with a small 1 mK temperature modulation during field sweeps at different fixed temperatures. The triangles mark the fields in which the constant normal state-specific heat value (dotted lines) is reached.

by the additional open circles, which is the only sharp feature that occurs. In the same field, a hysteretic opening of the two branches, which was recorded when the field was swept up and down, confirms the onset of superconductivity. At 350 mK $H_{c2}$

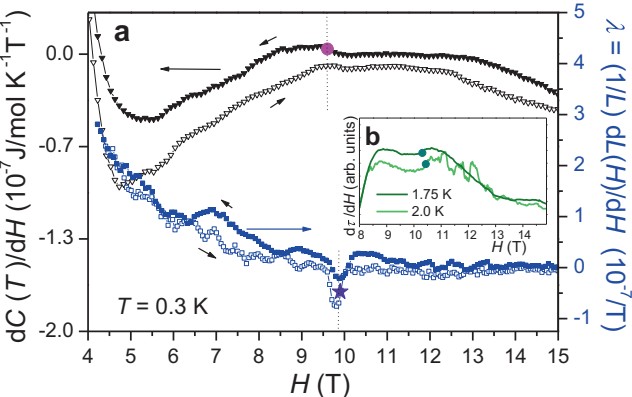

**Fig. 4 Thermodynamic bulk probes displaying the phase transition between the BCS low-field SC phase and the FLLO state in NbS$_2$. a** Magnetic field derivative d$C(T)$/d$H$ of the specific heat, measured at 300 mK during a field sweep (see Fig. 4**b** for the corresponding $C/T$ data) together with the linear magnetostriction coefficient $\lambda = (1/L)dL(H)/dH$ for magnetic fields applied strictly parallel to the layer structure. Both quantities reproducibly show a small transition anomaly at ~10 T, as marked by the circle and star symbols to be included in the phase diagram in Fig. 5. **b** Field derivative d$\tau(H)$/d$H$ of the magnetic torque for selected temperatures showing a small step-like anomaly near 10 T, as marked by the circles.

exceeds our highest field of 15 T, and a hysteresis exists up to 15 T (Fig. 2b).

In Fig. 2c, we present torque data measured in constant, strictly parallel fields as a function of temperature. The SC transition can be identified as a step-like transition. The data were normalized by the jump size at $T_c(H)$ for better clarity. We take the upper onset of the transition to identify $T_c(H)$ as marked by the additional open circles. Note that other criteria to define of $T_c(H)$ or $H_{c2}(T)$ provide qualitatively similar phase diagrams.

**Specific heat and thermal expansion experiments.** In Figs. 3 and 4, we summarize our specific heat and linear thermal expansion data. To achieve the parallel field orientation, we maximized the $T_c$ in a fixed field of a few Tesla by slightly rotating the sample in repeated measurements[24]. Figure 3a illustrates the specific heat $C/T$ and the linear thermal expansion coefficient $\alpha = (1/L)dL(T)/dT$ in a parallel field of 15 T. Both thermodynamic quantities, which are closely related via the thermodynamic Ehrenfest relationship, show a SC transition, which demonstrates that 15 T is not sufficient to suppress superconductivity. In the inset (Fig. 3b), we show specific heat data measured with our *ac* modulated temperature technique measured during field sweeps at constant temperatures of the thermal bath. A standard BCS superconductor would show the characteristic step-like transition at $H_{c2}$[36]. However, the NbS$_2$ data only display a broad bump centered at relatively low fields, while the signal fades very gradually towards higher fields to approach the normal state. There are no additional features to be seen here that could indicate additional phase transitions within the SC state. However, in Fig. 4a, we illustrate the magnetic field derivative of the 300 mK specific heat data together with the linear magnetostriction coefficient $\lambda = (1/L)dL(H)/dH$. For both quantities, a small anomaly indicates a phase transition within the SC state at ~10 T, which occurs at the theoretical Pauli limit. The transition is reproducible when the field is swept up and down. In the following, we will attribute it to the transition that separates the ordinary low-field SC state from a high-field FFLO state. In the magnetic torque $\tau(H)$ in the temperature range of 0.3–2.5 K tiny

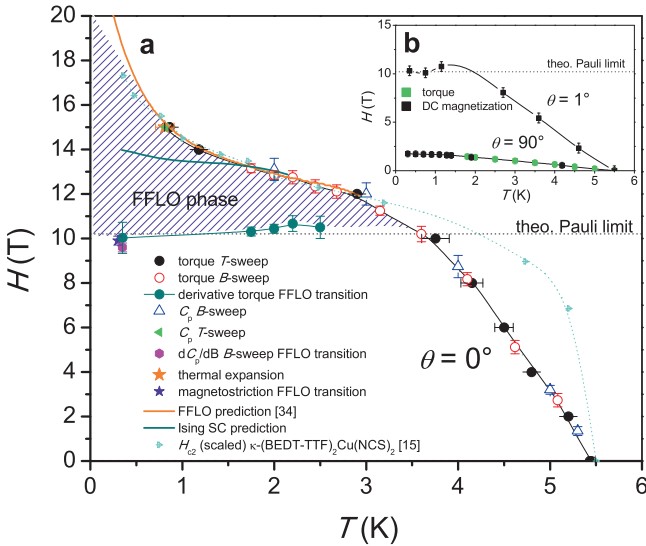

**Fig. 5 Magnetic field vs. temperature phase diagram of NbS$_2$ in magnetic field applied parallel to the layer structure. a** Filled black circles represent the upper critical field $H_{c2}(T)$ as measured by torque magnetometry during field scans at constant temperatures (B-sweep, Fig. 2a, b), red open circles represent the critical temperature $T_c(H)$ measured by torque during temperature scans (T-sweep, Fig. 2c). Blue triangles are $H_{c2}(T)$ data obtained from ac calorimetry measured during field scans ($C_p$ B-sweep, Fig. 3b). The orange star and green triangle mark the critical temperature in $C_p(T)$ ($C_p$ T-sweep) and thermal expansion $\alpha(T)$ data measured in 15 T (Fig. 3a). The violet star marks a small additional transition anomaly (FFLO transition) in the magnetostriction and filled magenta circle marks a similar anomaly visible in the field derivative of the specific heat near the Pauli limit due to the transition separating the ordinary low-field superconducting phase from the high-field FFLO state (Fig. 4a). In addition, the dark cyan circles correspond to a small anomaly in the magnetic torque attributed to the same phase transition within the superconducting state (Fig. 4b). The orange and dark cyan lines are theoretical predictions of the low-temperature $H_{c2}$ line for superconductivity with the Fulde–Ferrell–Larkin–Ovchinnikov (FFLO) state[34] and for Ising superconductivity, respectively. The FFLO phase is marked by the violet-shaded background. We have also added a scaled $H_{c2}$ curve of the organic superconductor κ-(BEDT-TTF)$_2$Cu(NCS)$_2$, which shows a very similar upturn of the $H_{c2}$ line due to the formation of an FFLO state[15]. Note that the characteristic temperatures/fields have been marked in the data shown in Figs. 1–4. **b** $H_{c2}$ data obtained from torque magnetometry in field scans measured with a small misalignment angle (θ = 1°, Fig. 1), for which no FFLO state is found, and for a field applied perpendicular to the layer structure (θ = 90°).

anomalies are hidden in the large slope near 10 T, but become visible after subtraction of a linear background in the form of a downward step, or as a dip-like structure in the field derivative $d\tau(H)/dH$ (Fig. 4b).

## Discussion

We summarize our results in the H/T phase diagram in Fig. 5. At high temperatures, the $H_{c2}$ transition line begins to rise with a non-vertical slope. This is in contrast to the behavior found in 2D samples and shows that there are significant orbital effects on the upper critical transition, which are naturally absent in 2D samples. We also include scaled $H_{c2}$ data of the organic superconductor κ-(BEDT-TTF)$_2$Cu(NCS)$_2$[15], for which numerous studies have demonstrated the existence of an FFLO state[13–15,18–21]. It is obvious that the initial slope of NbS$_2$ is much weaker than for κ-(BEDT-TTF)$_2$Cu(NCS)$_2$, which proves a

smaller Maki parameter $\alpha_m = \sqrt{2}\frac{H_{orb}(0)}{H_P(0)}$. A large Maki parameter in the range between 1.7 and 3.4[37,38] is considered one of the decisive prerequisites for the formation of an FFLO state. From a fit with the Werthamer–Helfand–Hohenberg model of the initial $H_{c2}$ slope of NbS$_2$ we obtain an orbital limit $H_{orb}$ ~23 T, which gives $\alpha_m = 3.25$, which is certainly large enough to support an FFLO state. Moreover, in quasi-2D superconductors with weak intra-layer coupling, the orbital effect is expected to be suppressed in high parallel magnetic fields when the electron wave functions become localized on the layers[39]. At 3.6 K, the $H_{c2}$ line rises above the theoretical BCS Pauli limit at ~10 T and then begins to saturate down to 1.5 K. At lower temperatures it shows a characteristic upswing and raises above 15 T, where it leaves the field region accessible in our experiment. The upswing is strikingly similar to that observed in κ-(BEDT-TTF)$_2$Cu(NCS)$_2$, with both data being almost congruent in the high-field region. Such an upswing indicates a certain change in the SC properties making it more robust against the strong Zeeman fields. It is generally interpreted as an indication of the development of an FFLO state. In fact, the phase diagram looks remarkably similar to other FFLO systems[10–24], including the field-induced transition at ~10 T, as determined from the data in Fig. 4, which probably indicates the phase change between the ordinary SC state at low fields and the high-field FFLO state. The fact that this transition occurs near the Pauli limit further supports the FFLO scenario, as the ordinary SC state should vanish beyond this field.

The inset of Fig. 5 shows $H_{c2}$ lines for magnetic fields with two different out-of-plane orientations (θ = 1° and 90°). It is obvious that the characteristic upturn is suppressed when the NbS$_2$ sample is misaligned by only 1° and remains absent for the field oriented perpendicular to the layers. This illustrates how the stronger orbital effects suppress the FFLO state even at small misalignment angles.

In most other FFLO systems, the $H_{c2}$ transition typically sharpens and becomes of first order as one approaches the Pauli limit[11,14,24]. In NbS$_2$, however, the $H_{c2}$ transition in high parallel fields remains very continuous, indicating a very gradual decay of superconductivity towards the normal state in the form of a second-order transition. In ref. [40], it was shown that the first-order nature of the $H_{c2}$ transition in high fields occurs only in the absence of the orbital effect, while in its presence it remains of second order anywhere. With orbital effect, it is expected that only the transition between the FFLO state and the BCS state should be of first-order nature. This is fully consistent with our observation: The finite $H_{c2}$ slope in the high-temperature range is clearly due to the orbital effect, and the additional transition at 10 T, observed as a spike in the magnetostriction coefficient, indicates a first-order nature of the FFLO to BCS transition. However, a small misalignment of only 1° sharpens the transition observed in torque to a step-like more first-order-like transition and indicates that superconductivity becomes Pauli limited. The fact that the $H_{c2}$ transition line then goes through a maximum is indeed expected for the Pauli-limited case in tilted fields[40].

ISOI is expected to exist predominantly in monolayers with their broken inversion symmetry leading to spin splitting of the Fermi surface, while in bilayers and the bulk it is spin degenerate[5]. However, at least bilayers still exhibit Ising superconductivity with $H_{c2}$ values exceeding the Pauli limit by a factor of ~4[5], which is a consequence of the very weak inter-layer coupling in TMD materials. It remains unclear at what thickness the Ising superconductivity disappears and whether Ising superconductivity can still have some influence in the bulk. Therefore, one concern regarding the FFLO scenario as an explanation of the phase diagram is that the ISOI could still have a considerable influence in the bulk. ISOI is not compatible with a finite-

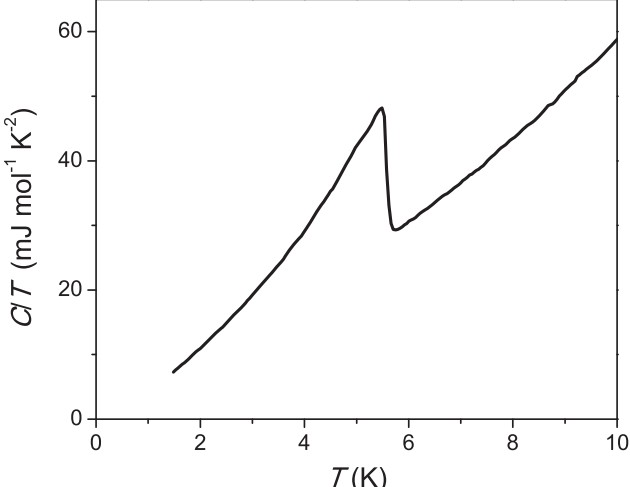

**Fig. 6 Zero-field specific heat $C/T$ of the NbS$_2$ single crystal used in this study.** A sharp superconducting transition is centered at $T_c = 5.5$ K.

momentum pair density wave state in the FFLO phase because it suppresses the effect of the in-plane Zeeman field on the Fermi surface. However, NbS$_2$ has multiple Fermi pockets at the $\Gamma$ and K/K' points. While the K pockets are protected by ISOI, the spin-orbit interaction at the $\Gamma$ pocket is weak and therefore compatible with the FFLO state. To test whether the phase diagram could alternatively be explained by Ising superconductivity, we calculated the critical field numerically by solving the linearized gap equation, using a three-band tight-binding model[41] with a spin-orbit coupling strength of $\beta_{SO} = 87$ meV (see "Methods" section for details). In order to simulate a bulk sample, we used a thickness of ten layers, which proved to be sufficient since we found that increasing the thickness further does not affect the critical field much. The obtained low-temperature $H_{c2}$ line is included in Fig. 5. While for samples in the 2D limit it could be shown that Ising superconductivity can cause very pronounced $H_{c2}$ upturns at low temperature similar to FFLO states[6,35,42], it is obvious that for reasonable values of the spin-orbit coupling strength the upturn of bulk NbS$_2$ is much too weak to explain our phase diagram.

To conclude, with the additional evidence of a field-driven phase transition within the SC state near the Pauli limit and the good agreement with theoretical simulations, it can be concluded that the phase diagram we observed for a bulk NbS$_2$ single crystal in magnetic fields strictly parallel to its layer structure, with its pronounced upswing of the $H_{c2}$ line far above the Pauli limit for superconductivity, is most plausibly explained by an FFLO state that forms in magnetic fields above 10 T. This high-field phase requires further systematic experiments including nuclear magnetic resonance to directly monitor the nature of the FFLO pair density wave state.

## Methods

**Sample preparation**. The growth of the high-quality 2H-NbS$_2$ single crystal was conducted in a quartz glass ampule, which was placed in a furnace with a temperature gradient, which serves to achieve oversaturation. The hot temperature was regulated in the range of 850–400 °C, while the cold side was held at 100–70 °C lower temperature. Upon gradually cooling the multicomponent flux through this temperature range the solubility of the components was reduced so that crystallization occurred. The feed of the chalcogenide was placed in the hot part of the reaction vessel, gradually dissolved into a growth medium in form of a CsCl–KCl–NaCl salt melt mixture of eutectic composition, and migrated to the cold end of the ampule, where it formed the crystals. The crystal growth lasted about 3 weeks. Detailed characterization of the process is found in ref. [43]. The thin platelets of high-quality single crystals of NbS$_2$ with optically flat surfaces on both sides were cut into a bar shape. Tablet-like crystals of NbS$_2$ grew up to 1 mm in size

when using an evaporation method[44]. The sample under investigation represented a small square-shaped platelet, which was completely flat on the macroscopic scale. The zero-field specific heat (Fig. 6) displays a sharp SC transition jump centered at $T_c = 5.5$ K indicative for a large SC volume fraction and good quality of the single crystal.

**Experimental techniques**. All our experimental sensors fit on the same piezo rotary stage mounted on our $^3$He probe in a 15 T magnet cryostat. The rotator allows the sample to be aligned in relation to the field direction with millidegree accuracy. Field sweeps were conducted at a rate of 0.1–0.5 T/min. Temperature-dependent measurements at different fixed fields were performed during sweeps at 0.04 K/min.

The magnetic torque $\tau$ was measured using a capacitive cantilever technique[24]. The cantilever leg is insulated from the counterplate of the capacitor by a thin sapphire sheet. This allows reversible measurements of the torque both as a function of the field or temperature. The capacitance is measured with a General Radio 1615-A capacitance bridge in combination with a SR830 lock-in amplifier. The vector quantity torque $\tau$ is directly related to the anisotropic DC magnetization by the relation $\boldsymbol{\tau} = \mathbf{M} \times \mathbf{H}$, where $\mathbf{H}$ is the applied magnetic field. Since for such layered superconductors the DC magnetization is expected to be greatest in the out-of-plane direction, this relationship suggests that $\tau$ vanishes in parallel fields. In reality, it reaches a minimum, but does not disappear completely during a complete field sweep due to higher-order quadrupole components[15,24]. The ultra-high resolution allows us to detect the tiny signal with a very good signal-to-noise ratio. The data are presented as the magnitude of the magnetic torque $\tau$, where the direction is perpendicular to the applied field.

The thermal expansion was measured with a miniature capacitance dilatometer[45], in which the sample is pressed with a screw mechanism against one of the plates of a parallel-plate capacitor suspended by a firm spring mechanism. A change in the sample length, induced either by a change in temperature (thermal expansion) or by the field (magnetostriction), changes the distance between the plates and thus causes small changes in the capacitance, which is measured in the same way as described for the torque. Thermal expansion is a bulk thermodynamic quantity closely related to specific heat and allows us to detect small changes in the sample length that occur during phase transitions.

The specific heat $C_p$ was measured using an alternating temperature (ac) technique with a mK temperature modulation amplitude[24]. To account for the relatively flat temperature dependence of the $H_{c2}$ line in the low-temperature/high magnetic field ($H/T$) regime, we performed the experiments at fixed base temperature during field sweeps. Note that the data measured in this somewhat unusual way still represent the thermal response of the sample with respect to a small temperature change, although here we probe the variation of the specific heat with respect to the field. The reason for this is that near the Pauli limit the slope of the $H_{c2}$ line in the magnetic field vs. temperature ($H$-$T$) phase diagram becomes very small and measurements during field sweep sampling reveal much more details in the important part of the phase diagram. The calorimeter platform was supported by thin nylon wires that become stiff at low temperature and serve as a support to prevent magnetic torque effects.

**Numerical simulation of the critical field to test for the influence of Ising superconductivity**. To test whether Ising superconductivity could account for the pronounced upturn of the $H_{c2}$ line in Fig. 5 at low temperature, we calculated the upper critical field numerically by solving the linearized gap equation, which can be written as

$$\frac{1}{g} = T \int_{BZ} d^2\mathbf{k} \sum_{n,l,m} \frac{|\langle l,\mathbf{k}|\delta\mathbf{H}_\Delta|m,-\mathbf{k}\rangle|^2}{[i\omega_n - \xi_l(\mathbf{k})][i\omega_n + \xi_m(-\mathbf{k})]} \quad (1)$$

where $\omega_n = \pi T(2n + 1)$ is the Matsubara frequency, $\xi_m(\mathbf{k})$ is the $m$-th eigen-energy of the normal Hamiltonian (including the Zeeman field) of wave vector $\mathbf{k}$, $|m,\mathbf{k}\rangle$ is the corresponding eigenstate, and $\mathbf{H}_\Delta$ is the matrix form of the pairing terms in the Hamiltonian

$$\delta\hat{\mathbf{H}}_\Delta = \sum_j \psi^\dagger_{j\uparrow}(\mathbf{k})\psi^\dagger_{j\downarrow}(-\mathbf{k}) \quad (2)$$

The index $j$ includes the layer index and the band index. With the number of layers being ten and the number of bands in each layer being 3, the dimension of the matrix $\delta\mathbf{H}_\Delta$ is 60 by 60 (with a factor of two from the spin). For a certain interaction strength $g$, Eq. (1) provides the relation between the temperature $T$ and the critical field $H_{c2}$. The actual $g$ is determined by the measured critical temperature $T_c$ at zero-field.

## Data availability

The experimental data supporting the findings of this work are available at https://doi.org/10.4121/14623011.v1.

## Code availability

The relevant codes needed to evaluate the findings of this study are available from the corresponding author upon reasonable request.

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

## Acknowledgements

We thank U. Lampe for technical support. This work was supported by grants from the Research Grants Council of the Hong Kong Special Administrative Region, China (GRF-16302018, GRF- 16303820, C6025-19G-A, SBI17SC14). M.A.-H. acknowledges the financial support from the Swedish Research Council (VR) under the project no. 2018-05393 and by the Arab-German Young Academy of Sciences and Humanities (AGYA). M.A.-H and D.C. acknowledge the financial support by the Megagrant 2020-220-08-6358 of the Government Council on Grants of the Russian Federation.

## Author contributions

This work was initiated by R.L.; C.w.C., J.L., C.Y.N., and K.T.L. carried out the magnetic torque experiments. C.Y.N., J.L., and R.L. carried out the specific heat experiments; the thermal expansion experiments were conducted by C.Y.N. and R.L, the DC magnetization measurements were conducted by K.T.L; the single crystal sample was provided by D.C, T.A.A.B., and M.A.H; J.J.H. provided the numerical simulations. The manuscript was prepared by R.L. and all authors were involved in discussions and contributed to the manuscript.

## Competing interests

The authors declare no competing interests.
