## [Peer Review File · Nature Communications]

REVIEWER COMMENTS

Reviewer #1 (Remarks to the Author):

I recommend the paper entitled "Superconductivity beyond Pauli's limit in bulk NbS₂: Evidence for the Fulde-Ferrell-Larkin-Ovchinnikov state" Chang-woo Cho et al. for publication.

However, I think the authors should consider the following revisions:

1)

The main evidence for an FFLO state comes from an upturn in the experimental H_{c2} curve shown in Figure 5. This is then compared to a prediction for a model for Ising superconductivity (solid green curve). In my opinion, the paper does not give enough detail on how the Ising limit was calculated. Ref. 38 only explains tight binding band structure calculations for dichalcogenides. I suggest adding more details on how to go from Ref. 38 to the solid green line of Fig. 5.

2)

I fail to see the relevance of the mauve crosses and teal solid circles. I can see that these curves level off at the Pauli limit. However, is there a particular reason this should happen? I suggest to give some explanation.

3)

Open circles in Fig. 1, and the orange stars in Fig. 4b for magnetostriction. I believe some discussion on how the position of the phase transition was determined from these measurements is appropriate, as the transitions are rather subtle.

4)

Abstract "We present magnetic torque, specific heat and thermal expansion measurements combined with a piezo rotary positioner of the bulk transition metal dichalcogenide (TMD) superconductor NbS₂ in high magnetic fields applied strictly parallel to its layer structure. The upper critical field of superconducting TMDs in the 2D form is known to be dramatically enhanced by a special form of Ising spin orbit coupling. This Ising superconductivity is very robust against the Pauli limit for superconductivity."

The phrase with the piezo rotary positioner requires an explanation on why such a device is needed. The last phrase needs to be rewritten.

5)

"Among them are intrinsic superconductors with 2H-NbSe₂ and 2H-NbS₂ as representatives of the highest critical temperatures."

With respect to what are the T_c compared?

6)

"In their 2D form, they have aroused great interest due to the discovery of Ising superconductivity, which allows them to withstand the Pauli limit of superconductivity [4,5]."

Withstand should probably be replaced with exceed.

7)

"In reality bilayers have H_{c2} values, which still exceed the Pauli limit by a factor of ~ 4 [5]."

From the way this sentence is written it is not clear if it is good or bad that H_{c2} exceeds the Pauli limit of ISOI superconductors.

Reviewer #2 (Remarks to the Author):

The present manuscript describes magnetic torque, specific heat and thermal expansion measurements on single crystalline NbS₂. The results suggest an enhancement of the upper critical field H_{c2} above the so-called Pauli limit which the authors attribute to the presence of the Fulde-Ferrell-Larkin-Ovchinnikov (FFLO) state.

The FFLO state is an exotic form of superconductivity and as such, evidence for such a state has been must sought after by experimentalists. The number of cases for which this evidence is considered robust, however, is rather small and as such, one should tread carefully when considering candidate materials. Provided that evidence is robust, however, the more experimental realizations of the FFLO state are made, the more opportunities there will be to study this exotic state.

In my opinion, the evidence presented in the current manuscript is reasonably convincing and therefore, I am tempted to recommend its publication. Prior to that, however, I suggest the authors seriously consider the following additions to their manuscript:

(i) Comparative measurements for H/c , in particular of the T-dependence of H_{c2}, to confirm whether or not the upturn is observed in this field orientation. As stated by the authors, NbS₂ is a multi-pocket conductor. As such, there is always the possibility of that the upturn in H_{c2} is a signature of multi-band superconductivity. Measurements in the orthogonal direction should help address whether or not the upturn reported in this manuscript is solely induced by in-plane magnetic fields.

(ii) One of the criterion for the observation of FFLO state is a that the superconductor is in the ultra-clean limit, meaning that the mean free path is significantly longer than the corresponding coherence length. Discussion of this point should also be included in the present manuscript.

Provided these two aspects are addressed, I would be happy to consider its publication in Nature Communications.

I note, in passing, that the most recent evidence for the FFLO state was reported in FeSe by Kasahara et al. (PRL vol. 124, 107001 (2020)) not in KFe₂As₂.

Responses to Reviewer #1

Comment: *I recommend the paper entitled “Superconductivity beyond Pauli’s limit in bulk NbS₂: Evidence for the Fulde-Ferrell-Larkin Ovchinnikov state” Chang-woo Cho et al. for publication.*

Our reply: We thank the reviewer for the positive evaluation!

Comment: *The main evidence for an FFLO state comes from an upturn in the experimental H_{c2} curve shown in Figure 5. This is then compared to a prediction for a model for Ising superconductivity (solid green curve). In my opinion, the paper does not give enough detail on how the Ising limit was calculated. Ref. 38 only explains tight binding band structure calculations for dichalcogenides. I suggest adding more details on how to go from Ref. 38 to the solid green line of Fig. 5.*

Our reply: The green curve of H_{c2} in Fig. 5 is obtained by numerically solving the linearized gap equation, which can be written as

$$\frac{1}{g} = T \int_{BZ} d^2\mathbf{k} \sum_{n,l,m} \frac{|\langle l, \mathbf{k} | \delta H_{\Delta} | m, -\mathbf{k} \rangle|^2}{[i\omega_n - \xi_l(\mathbf{k})][i\omega_n + \xi_m(-\mathbf{k})]}. \quad (R1)$$

where $\omega_n = \pi T(2n + 1)$ is the Matsubara frequency, $\xi_m(\mathbf{k})$ is the m -th eigenenergy of the normal Hamiltonian (including the Zeeman field) of wave vector \mathbf{k} with $|m, \mathbf{k}\rangle$ being the corresponding eigenstate, and H_{Δ} is the matrix form of the pairing terms in the Hamiltonian

$$\delta \hat{H}_{\Delta} = \sum_j \psi_{j\uparrow}^{\dagger}(\mathbf{k}) \psi_{j\downarrow}^{\dagger}(-\mathbf{k}).$$

The index j includes the layer index and the band index. With the number of layers being ten and the number of bands in each layer being 3, the dimension of the matrix δH_{Δ} is 60 by 60 (with a factor of two from the spin). For a certain interaction strength g , Eq. (R1) gives the relation between the temperature T and the critical field H_{c2} . The actual g is determined by the measured critical temperature T_c at zero field.

We have added these details to the Methods section (page 7, from line 10).

Comment: *I fail to see the relevance of the mauve crosses and teal solid circles. I can see that these curves level off at the Pauli limit. However, is there a particular reason this should happen? I suggest to give some explanation.*

Our reply: The hallmark of the FFLO state is not only the H_{c2} upturn at low temperatures, in addition a phase transition should occur that separates the ordinary low-field superconducting phase from the high-field FFLO state. This transition is expected to occur not far from the line representing the Pauli limit, although the exact position may vary depending on the material parameters. The data mentioned highlight features that indicate this phase transition within the superconducting state. It indeed occurs near the Pauli limit. To highlight this more clearly, we have marked the regime of the FFLO state in the phase diagram (revised Fig. 5.) with a magenta shaded background. We have also simplified the diagram by focusing on the exact parallel orientation, while data in tilted fields have been moved to an inset.

The mauve star together with the magenta hexagon are taken from field sweep data of the magnetostriction and specific heat showing a small phase transition anomaly at about 10 T at $T=0.3$ K. The corresponding data are shown in Fig. 4a and we have inserted the same symbols into these revised graphs to highlight the transition field in Fig. 4 (see also page 3, line 39-42). The teal-colored solid circles are from anomalies in the magnetic torque, and a selection of corresponding data is shown in Fig. 4b. Again, we have used the same symbols as in Fig. 5 in the revised plots to mark the anomalies in Fig. 4b. Another curve (magenta crosses) in the plot results from the torque data measured in a field tilted by 1 degree, as shown in Fig. 1, which also flattens out near the Pauli limit. This implies that the FFLO state is no longer formed in such a tilted field and the data remain Pauli limited. In the revised manuscript we explain this more clearly (page 4, line 30-34 and caption of Fig. 5).

Comment: *Open circles in Fig. 1, and the orange stars in Fig. 4b for magnetostriction. I believe some discussion on how the position of the phase transition was determined from these measurements is appropriate, as the transitions are rather subtle.*

Our reply: Concerning Fig. 1: For this tilted field data, there is in principle only one phase transition occurring at H_{c2} , and the steep step-like transition in the torque likely indicates a first-order Pauli-limited nature. We therefore decided to use the points where the steepest slope occurs near H_{c2} , as mentioned in the caption, which should best represent the first-order transition. Note that other criteria (e.g. the very continuous small transition onset) only lead to an almost parallel shift of the phase transition line. In the revised manuscript we explain more clearly why we use this criterion (page 3, line 1-4). This is different from the exactly parallel alignment, where the much sharper kink-like onset (see Fig. 2) is the only sharp feature that occurs.

Regarding Fig. 4b, there is no orange star in Fig. 4b and the data in this graph represents torque data and not magnetostriction. The reviewer is probably referring to the orange star in Fig. 5, which was determined from Fig. 3. It represents thermal expansion data measured during a temperature sweep. We believe this transition is fairly sharply defined, and we used the midpoint of the jump to determine the transition temperature. We have added the orange star in Fig. 3 (as well as the magenta hexagon for C_p) to mark the transition temperatures, hopefully clarifying the origin of this symbol in Fig. 5.

Comment: *Abstract “We present magnetic torque, specific heat and thermal expansion measurements combined with a piezo rotary positioner of the bulk transition metal dichalcogenide (TMD) superconductor NbS₂ in high magnetic fields applied strictly parallel to its layer structure. The upper critical field of superconducting TMDs in the 2D form is known to be dramatically enhanced by a special form of Ising spin orbit coupling. This Ising superconductivity is very robust against the Pauli limit for superconductivity.”*

The phrase with the piezo rotary positioner requires an explanation on why such a device is needed. The last phrase needs to be rewritten.

Our reply: We thank the reviewer for pointing out our unclear wording in the abstract. Observing the FFLO state in layered compounds often requires strictly parallel fields, and, as can be seen from our data in Fig. 1, even 1 degree of misalignment causes superconductivity to no longer persist beyond the Pauli limit. The piezo rotator is therefore essential to align the layered structure of the sample with the field prior to the measurements. We have rephrased the abstract as follows: “We present measurements of

the magnetic torque, specific heat and thermal expansion of the bulk transition metal dichalcogenide (TMD) superconductor NbS₂ in high magnetic fields, with its layer structure aligned strictly parallel to the field using a piezo rotary positioner. ... superconductivity beyond the Pauli limit still exists in bulk single crystals of NbS₂ for a precisely parallel field alignment.”

The other sentence mentioned by the reviewer has been rewritten as: "This Ising superconductivity is very robust to the Pauli paramagnetic effect and can therefore exist beyond the Pauli limit for superconductivity."

Comment: *“Among them are intrinsic superconductors with 2H-NbSe2 and 2H-NbS2 as representatives of the highest critical temperatures.” With respect to what are the Tc compared?*

Our reply: The intrinsic transition metal superconductors are mainly NbSe₂, NbS₂ and TaS₂, with TaS₂ having a much lower T_c. We have removed the second half of the sentence, which now simply reads like: “Among them are intrinsic superconductors including 2H-NbSe₂ and 2H-NbS₂” (page 1, line 25).

Comment: *“In their 2D form, they have aroused great interest due to the discovery of Ising superconductivity, which allows them to withstand the Pauli limit of superconductivity [4,5].” Withstand should probably be replaced with exceed.*

Our reply: We thank the reviewer for this suggestion, we have replaced it accordingly.

Comment:

“In reality bilayers have H_{c2} values, which still exceed the Pauli limit by a factor of ~4 [5].” From the way this sentence is written it is not clear if it is good or bad that H_{c2} exceeds the Pauli limit of ISOI superconductors.

Our reply: It was not our intention to make a statement about whether it is good or bad. What we wanted to express is that Ising superconductivity is not limited to monolayers but still occurs in multiple layers, and it remains unclear at what thickness it is suppressed. In this sense, it is not clear whether Ising superconductivity could still exist to some extent in the limit of bulk samples. We have rewritten this sentence to express this more clearly (page 5, line 10 - 15).

Responses to Reviewer #2

Comment: *In my opinion, the evidence presented in the current manuscript is reasonably convincing and therefore, I am tempted to recommend its publication. Prior to that, however, I suggest the authors seriously consider the following additions to their manuscript:*

Our reply: We thank the reviewer for the positive review!

Comment: *Comparative measurements for $H//c$, in particular of the T -dependence of H_{c2} , to confirm whether or not the upturn is observed in this field orientation. As stated by the authors, NbS₂ is a multi-pocket conductor. As such, there is always the possibility of that the upturn in H_{c2} is a signature of multi-band superconductivity. Measurements in the orthogonal direction should help address whether or not the upturn reported in this manuscript is solely induced by in-plane magnetic fields.*

Our reply: We did not focus on perpendicular fields, as such data are available in the literature (e.g. PRB 82, 014518 (2010)), but we quickly performed such measurements with both SQUID and torque magnetometry. The data are included in the inset of revised Fig. 5 of the manuscript. These data show no upturn at low temperatures, which we comment on in the revised manuscript (page 4, line 30-34).

Comment: *One of the criterion for the observation of FFLO state is a that the superconductor is in the ultra-clean limit, meaning that the mean free path is significantly longer than the corresponding coherence length. Discussion of this point should also be included in the present manuscript.*

Our reply: We have included this statement, along with a reference, in the revised manuscript (page2, line 9 - 12).

Comment: *I note, in passing, that the most recent evidence for the FFLO state was reported in FeSe by Kasahara et al. (PRL vol. 124, 107001 (2020)) not in KFe₂As₂.*

Our reply: We thank the reviewer, yes of course this work should be cited and we have added the reference (page 2, line 9).

REVIEWERS' COMMENTS

Reviewer #1 (Remarks to the Author):

I find the authors have addressed all the points of my previous review and the manuscript should be published.

Reviewer #2 (Remarks to the Author):

I have looked at the response of the authors to the previous referee reports and the revised manuscript and I am satisfied that they have addressed the main concerns of the referees and have improved the clarity of the manuscript accordingly. I am therefore happy to recommend publication of the paper in its present format. It is remarkable to me how the original predictions of Fulde, Ferrell, Larkin and Ovchinnikov have come to be realized in such diverse families of superconductors. This manuscript adds the dichalcogenides to that list.